# Indirect Neonatal Hyperbilirubinemia and the Role of Fenofibrate as an Adjuvant to Phototherapy

**DOI:** 10.3390/children10071192

**Published:** 2023-07-10

**Authors:** Salam K. Shabo, Khalaf H. Gargary, Omer Erdeve

**Affiliations:** 1Heevi Paediatrics Teaching Hospital, Duhok 42001, Iraq; 2College of Medicine, University of Duhok, Duhok 42001, Iraq; 3Department of Pediatrics, Division of Neonatology, Faculty of Medicine, Ankara University, Ankara 06830, Türkiye

**Keywords:** neonatal hyperbilirubinemia, fenofibrate, phototherapy, serum bilirubin

## Abstract

Background: One of the most prevalent illnesses in neonates that needs care and treatment is neonatal jaundice. Several drugs are used as pharmacological modalities for treating hyperbilirubinemia, like intravenous immunoglobulin, D-penicillamine, metalloporphyrin, phenobarbital, zinc sulfate and clofibrate. Previous studies suggest the usefulness of fenofibrate in the treatment of hyperbilirubinemia. Objectives: The study aims at assessing the effectiveness of oral fenofibrate in the treatment of indirect neonatal hyperbilirubinemia in full-term neonates. Method: This is a quasi-experimental study that was conducted at Heevi Pediatrics Teaching Hospital in Duhok, which is located in the Kurdistan Region of Iraq. It involved term infants who had jaundice. The neonates who were eligible for the study were randomly assigned to one of two groups: the intervention group or the control group. Both groups were treated with conventional phototherapy. Fenofibrate was administered in a single oral dose of 10 mg/kg to the participants in the intervention group. Throughout the entirety of the treatment, levels of total serum bilirubin were compared and contrasted between the two groups. Results: After 12 h of treatment, a statistically significant difference (*p*-value = 0.001) was seen in the serum bilirubin levels between the two groups. The difference in serum bilirubin levels became significantly progressively pronounced after 24, 48, and 72 h. The average time of discharge was 63.6 h for the intervention group and 90.9 h for the control group, and this difference was statistically significant (*p*-value < 0.001). Conclusions: The time it takes to lower high bilirubin levels in neonates may be shortened by combining conventional phototherapy with a single oral dosage of 10 mg/kg fenofibrate. Consequently, these neonates will experience a shorter hospitalization and an accelerated discharge from the hospital.

## 1. Introduction 

One of the most frequent illnesses in newborns worldwide that requires immediate attention and treatment is neonatal jaundice. During the first week of life, jaundice develops in about 60% of term babies and 80% of preterm babies. Physiological neonatal jaundice is harmless; however, jaundice in a newborn is a constant source of concern for parents due to the dread of bilirubin toxicity in the developing brain and immature organs [1].

Unconjugated bilirubin, also known as indirect bilirubin or UCB, is a natural antioxidant that functions at low concentrations but can be harmful to a growing brain. Its toxicity in otherwise healthy full-term infants and premature infants is highly debatable, and it is still not known which level of bilirubin is regarded as safe in infants, especially those who were born prematurely and/or with a low birth weight. However, the levels that are toxic to one neonate may not be toxic to another one, or even to the same infant under different clinical conditions [2].

A high level of unconjugated bilirubin may result in acute and/or chronic serious complications, such as severe occipital and temporal lobe seizures as well as kernicterus, also known as bilirubin encephalopathy, which is permanent brain damage involving a wide range of symptoms, such as tone abnormalities, feeding difficulty, aberrant crying, and kernicteric facies, that is graded by the bilirubin-induced neurological dysfunction (BIND) score or modified BIND [3]. The challenges are especially acute in low- and middle-income countries, where the burden of the disease is greatest and medical facilities are least equipped to manage severe neonatal jaundice [4].

Due to the rapid adaptation of bilirubin metabolism that takes place immediately after birth, infants produce more bilirubin than adults. As a direct result of this, newborn red blood cells have a quicker turnover rate and a shorter lifespan than adult red blood cells. Other factors, such as increased liver cell bilirubin production (which can be caused by hemolytic anemias, polycythemia, bruising or internal hemorrhage, increased enterohepatic circulation, or infection), decreased activity of the liver transferase enzymes, enzyme blockers, and decreased bilirubin uptake by liver cells, are also potential contributors to unconjugated hyperbilirubinemia in neonates [5].

The higher bilirubin production and the limitations of conjugating it lead to physiologic jaundice. This is a condition with high serum bilirubin concentrations in the first days of life in full-term infants and up to the first week in preterm infants. When the serum bilirubin values exceed the threshold of 7 to 17 mg/dL, the condition is referred to as exaggerated physiologic jaundice [6,7].

There are several non-pharmacological and pharmacological modalities for treating hyperbilirubinemia. Phototherapy is considered the most widely used non-pharmacological therapy for the treatment and prophylaxis of neonatal unconjugated hyperbilirubinemia [8]. This method is based on the concept that absorption of light through the skin converts unconjugated bilirubin into bilirubin photoproducts that are excreted in the stool and urine. The effectiveness of phototherapy in declining the rates of total serum bilirubin (TSB) is variable, but a 6% to 20% decrease is expected [9].

In spite of phototherapy’s extensive use in the treatment of unconjugated neonatal hyperbilirubinemia, the method is associated with a number of problems, some of which are short-term and others of which are long-term. These include the requirement that, throughout the process, the mother and the newborn be physically separated from one another, as well as the need to draw blood from the infant. In some cases, the procedure may also result in the patient having to remain in the hospital for a longer period of time, experiencing heat, having loose stools, becoming dehydrated, and perhaps causing damage to the retina. There is a decrease in cellular viability and an increase in cellular apoptosis that occurs in vitro when human retinal pigment epithelial cells (HRPEpiC) are subjected to three light–darkness cycles (12 h/12 h) of radiation [10].

The most significant suggested treatments for hyperbilirubinemia are phototherapy and exchange transfusion, but with recent advancements in therapy, the usage of numerous medications has increased [11,12]. Intravenous immunoglobulin, D-penicillamine, metalloporphyrin, phenobarbital, zinc sulfate, and clofibrate are some of the medications used [13,14].

Fibrates, which are a derivative of phenoxy-isobutyric acid, have been used for many years as hypolipidemic medications; they may also be used to treat hyperbilirubinemia. Fenofibrate assists in the treatment of neonatal hyperbilirubinemia through various mechanisms, which generally involve the upregulation of enzymes involved in bilirubin metabolism and excretion. By activating peroxisome proliferator-activated receptor alpha (PPAR-α), fenofibrate promotes the expression of uridine diphosphate glucuronosyltransferase (UGT) enzymes, which aid in the conjugation and elimination of bilirubin. Furthermore, fenofibrate boosts the expression of organic anion transporters, which facilitates the excretion of conjugated bilirubin into neonates’ bile [15].

Fenofibrate has a half-life that ranges between 18 and 20 h. Good absorption of the medicine occurs in the gastrointestinal tract, and the vast majority of the drug is eliminated in the urine, where it is largely excreted as fenofibric acid and fenofibric acid glucuronide [16]. Clofibrate has been utilized in the majority of investigations looking at how fibrates affect hyperbilirubinemia. However, side effects, including nausea, limit the usefulness of this medication [17].

Fenofibrate is most similar to clofibrate in its mechanism of action. It is more tolerable, and therefore safer, in the pediatric age group. In a different trial, the drug’s safety was shown to have been proven, and after a single dose as well as during follow-up for a period of six months, no adverse effects were noticed. This was despite the fact that the dose used in the other study was significantly larger than the dose used in our study (100 mg/kg of fenofibrate) [16]. On the other hand, many studies examined the use of fenofibrate in the treatment of neonatal hyperbilirubinemia and demonstrated its usefulness. Oral administration of fenofibrate as a single dose of 10 mg/kg combined with phototherapy could reduce the total serum bilirubin (TSB) and the duration of hospitalization [8,17,18,19,20,21,22,23].

In the region where this trial was conducted, the use of fenofibrate as a treatment for newborn hyperbilirubinemia is not yet an established routine aspect of the treatment protocol. Because specific studies on the use of fenofibrate in this region have not been completed, a regional study has the potential to demonstrate the medication’s effectiveness in our local community. The purpose of this research was to determine whether or not oral fenofibrate is successful in the treatment of indirect neonatal hyperbilirubinemia in neonates who were born full-term.

## 2. Patients and Methods

This study was a quasi-experimental study that was carried out at the Heevi Pediatrics Teaching Hospital in Duhok, which is located in the Kurdistan region of Iraq, during the months of August 2022 and February 2023. In partnership with the University of Duhok, the research ethics committee of the Directorate General of Health in Duhok granted permission for the study (reference number 18052022-3-4 on 18 May 2022).

### 2.1. Subjects

The participants in our study were full-term newborns who were diagnosed with unconjugated hyperbilirubinemia on their 2nd to 7th day of life, when they were brought to the hospital for evaluation. Furthermore, in order for the neonate to be eligible for participation in the research, there were a number of other critical factors that needed to be taken into consideration. These factors included a gestational age ranging from 37 to 41 weeks and a birth weight ranging from (2500 to 3500) g. In addition to this, the total serum bilirubin TSB needed to be between 15 and 20 mg/dL, and the reason for hyperbilirubinemia needed to be either ABO or Rh incompatibility, exaggerated physiological jaundice, or both. The following factors were considered incompatible with participation in the study:Preterm (<37 weeks gestational age).Conjugated bilirubin levels above 2 mg/dL or more than 15% of TSB.Congenital anomalies.Sepsis or exchange transfusion.Respiratory distress.Cephalohematoma or subgaleal bleeding.

### 2.2. The Study Groups

The research involved a total of 100 newborns who had reached their full gestation, split evenly between two groups, A and B, of 50 infants each. The size of the sample was established by taking into account the admission rate, the constraints placed on the duration of the study, and the estimates obtained from earlier research that was analogous. The following is a breakdown of the two groups:Group A—the intervention group consisted of 50 newborns who were given conventional phototherapy in addition to a single dose of oral fenofibrate suspension administered at a dose of 10 mg/kg.Group B—standard phototherapy was the sole treatment that any of the 50 newborns in the control group received.The neonates were not assigned to the different study groups in a random fashion. We considered a group of newborns who were only given conventional phototherapy to be a control when that group’s characteristics and measurements were almost identical to those of the intervention group’s newborns who were given the same number of assignments to be in that group. This procedure was repeated until there were fifty newborns in each of the groups.Guidelines established by the American Academy of Pediatrics (AAP) for term and near-term infants have been centered on when phototherapy should begin and end [24]. It was determined that a BT-400 (Korea) Phototherapy device would be utilized. This device is a floor-standing, mobile phototherapy light that produces a narrow band of high-intensity blue light via blue light-emitting diodes (LEDs). Blue LEDs produce light with a wavelength that falls between (400 and 550) nanometers (the peak wavelength falls between 450 and 475 nanometers), and it is anticipated that the light will continue to function as intended for roughly 20,000 h. This spectrum, which corresponds to the spectral absorption of light by bilirubin, is therefore regarded as the most efficient and secure for the destruction of bilirubin. The baby’s vital signs were monitored throughout the process, and a gap of around 40 cm was maintained between the newborn and the photo lamp. Eye pads and diapers were used to protect both the eyes and the genital area, respectively.

### 2.3. Data Collection

After being presented with all of the relevant information, the parents first provided their approval. Next, following that, in order to include or exclude neonates in the study, we collected demographic data such as (gestational age, birth weight, gender, and age in days of the neonate). In addition, a thorough history and physical examination were carried out in order to eliminate the probability of specific disorders, including sepsis, premature birth, respiratory distress, subgaleal hemorrhage, and cephalohematoma. Additionally, a blood sample was collected from both the intervention and the control groups in order to perform the following tests: the complete blood count (CBC), the total bilirubin (direct and indirect), the Coombs test, and the blood group (ABO and Rh of neonates and their mothers). Following the initiation of treatment, the total serum bilirubin of the neonates in both groups (A and B) was monitored every 12 h until they were deemed healthy enough to be discharged from the hospital.

### 2.4. Statistical Analysis

SPSS statistical software, version 26, was utilized in order to perform the analysis on the collected data. When presenting numerical data, the mean and standard deviation (SD) are the most common methods, although frequency tables are typically used when presenting categorical information. A Student independent t-test was used to estimate the primary outcome measure, which aimed to determine whether or not there were statistically significant differences in the mean levels of total serum bilirubin (TSB) between the intervention group and the control group. The Chi-squared test was applied in order to investigate any other category data-based distinctions that existed between the two groups. It was determined that the hypothesis was statistically significant if the *p*-value was less than 0.05.

## 3. Results

The research was carried out on a total of 100 neonates that were born at full-term. These infants were between their second and seventh days of life, with a mean age of 3.74 days in the group that received the intervention (fenofibrate) and 3.88 days in the group that served as the control. They were between 37 and 41 weeks pregnant, with a mean gestational age of 38.6 weeks in the intervention group and 38.5 weeks in the control group, respectively. In the intervention group, the average weight was 3.01 kg, while in the control group, the average weight was 3.02 kg. When the means of the two groups were examined using the independent t-test, as shown in Table 1, these seemingly insignificant changes in the numerical variables of age, gestational age, and weight between the two groups were not found to be statistically significant.

In terms of gender, the intervention group contained 28 male neonates (56%), while the control group contained 29 male neonates (58%). Both groups had the same number of male neonates overall. Three categories were considered when determining the cause of hyperbilirubinemia in the study participants: ABO incompatibility, Rh incompatibility, and exaggerated physiological jaundice. In total, 25 (50%) of the neonates in the group that received fenofibrate had an ABO incompatibility, and 15 (30%) had an Rh incompatibility. In the control group, the incompatibility rates for ABO and Rh blood types were 28 (56%) and 13 (26%), respectively. At the onset of the study, a direct Coombs test was administered to each newborn; the results were negative for 34 (68%) of the neonates in the intervention group and 37 (74%) of the neonates in the control group. Table 2 displays the results of a reevaluation of the categorical variables of gender, diagnosis, and Coombs test using the Chi-square test. There were no statistically significant differences between the groups revealed by the assays.

The most important finding from the research, which compared the two groups in terms of the rate of TSB reduction over time, is summarized in Table 3, and its graphical representation can be found in Figure 1. On admission, the mean TSB value in mg/dL in the intervention and control groups was 19.28 and 19.21, respectively, and was considered statistically not significant. Early in the course of treatment, after 12 h, a significant difference (*p*-value = 0.001) was seen in the serum bilirubin between the two groups. The mean was 17.92 in the intervention group and 18.65 in the control group, measured in mg/dL.

According to Table 3, there was a growing disparity in the TSB between the two groups after each of the following time periods:At 24 h, the mean TSB in the group that received treatment was 16.42 mg/dL, while the mean TSB in the control group was 17.69 mg/dL.At 48 h, the mean TSB value in the group that received fenofibrate was 14.81 mg/dL, while the value in the control group was 16.51 mg/dL.Finally, at 72 h, the mean TSB value in the group that was given fenofibrate was 12.97 mg/dL, while the value in the control group was 15.01 mg/dL.

In each and every one of these occurrences, the *p*-value for the difference was lower than 0.001, indicating that it was significant from a statistical point of view.

Most of the neonates in the control group were discharged from the hospital after 96 h, while the majority of the neonates in the intervention group were discharged from the hospital before 72 h. As a conclusion, it can be stated that the average amount of time that patients in the intervention group spent in the hospital was 63.6 h, whereas the average amount of time that patients in the control group spent in the hospital was 90.9 h. This disparity between the two groups suggests that the difference was statistically significant.

## 4. Discussion

In this study, we compared the effectiveness of treating hyperbilirubinemia in neonates with fenofibrate plus phototherapy to the effectiveness of treating the condition with phototherapy alone in the intervention group and the control group, respectively. Fenofibrate was given as a 10 mg/kg single oral dose. The design of the study was quasi-experimental, with a nonrandom allocation of neonates to the groups.

A comparison was carried out between the intervention and the control groups in order to ensure that the two groups were virtually identical with regard to general, clinical, and laboratory characteristics. None of these characteristics, including age, gestational age, weight, gender, diagnosis, Coombs test, or TSB on admission, showed a statistically significant difference between the two groups. A wide range of studies that have been conducted on the same subject, particularly randomized clinical trials, have examined the random distribution of the characteristics between the two groups and discovered that there are no significant variations between the study groups with regard to the inclusion characteristics [8,19,21].

After just 12 h of therapy, there was already a discernible gap in the rate of TSB improvement between the intervention and control groups. This is considered early when compared with other studies. Recently, Iranian researchers led by Saadat et al. published the results of a study in which they found that the total serum bilirubin levels at 48 h (*p* < 0.001) and upon discharge (*p* < 0.001) were significantly lower in the fenofibrate group than in the control group. The comparison in the length of hospital stays between the fenofibrate and control group was not statistically significant (*p* = 0.612), despite the fact that the fenofibrate group had shorter hospital stays [25].

Islam et al. [5] directed one of the two investigations that were carried out in Bangladesh, while Musharraf et al. [26] were in charge of the other investigation that was carried out there. These researchers looked into the relationship between a single dose of fenofibrate weighing 10 mg/kg and the severity of indirect newborn hyperbilirubinemia. The levels of bilirubin in the blood that were measured at the beginning of the first study were as follows: 16.86 mg/dL for the intervention group and 17.31 mg/dL for the control group. After 24 and 48 h of initiating phototherapy, the bilirubin levels in the intervention group declined to lower values of 14.83 mg/dL and 12.85 mg/dL, while the bilirubin levels in the control group stayed at 15.73 mg/dL and 14.20 mg/dL over those same time periods. The intervention group received a total of 51.20 h of phototherapy, whereas the control group received a total of 70.40 h. The second trial found that after 48 h, there was a significant drop in blood bilirubin as well as a shorter length of time spent in the hospital.

Chaudhary et al. performed a clinical trial on fenofibrate and hyperbilirubinemia in north India, and found that after 24 h, the fenofibrate and control groups still did not differ significantly. The difference became statistically significant after 36 h of treatment [17]. Two of the studies conducted in Egypt also did not find significant differences at 24 h and seen after that [19,21]. The early significant difference seen in this study is a good clue for the effectiveness of fenofibrate in treating neonatal jaundice in the study locality. The heterogeneity could be related to the fact that different studies may use various protocols for administering phototherapy, including phototherapy parameters such as light intensity, wavelength, treatment duration, and the type of phototherapy device. Moreover, investigations conducted on populations with diverse demographic and ethnic characteristics, the method employed to assess bilirubin, the precision and time of bilirubin measurement, and different inclusion and exclusion criteria among studies can significantly influence the interpretation of the findings and contribute to heterogeneity.

With daily checks of serum bilirubin in both groups, the study showed a gradual decline in TSB for all participants and a more rapid one in the intervention group. The difference remained statistically significant at 24, 48, and 72 h of treatment. Ahmadpour-Kacho et al. conducted a clinical trial in Iran in order to determine the impact of oral fenofibrate on total serum bilirubin in term infants with hyperbilirubinemia. The TSB was found to be 13.29 mg/dL on the second day of hospitalization in the intervention group and 14.06 mg/dL in the control group (*p* = 0.04). On the third day, it was found to be 9.99 in the intervention group and 11.01 in the control group (*p* = 0.006) [18]. These results were comparable to our study results in regard to the difference between the intervention and control groups, though the values were even lower.

There is an overall agreement in most of the studies that a single dose of 10 mg/kg fenofibrate combined with conventional phototherapy has a significant effect on lowering TSB in a shorter time compared with phototherapy alone [8,17,19,20,21,22,23]. In contrast, the research that Prabha and Saravanan [27] conducted found no statistically significant differences in the bilirubin reduction rate with a single dosage of fenofibrate (10 mg/kg) after 48 h in the intervention group. This was the case despite the fact that phototherapy was administered for a shorter period of time. Given the frequency with which this finding appears in the existing body of literature, the conclusion that fenofibrate does not have a substantial effect on hyperbilirubinemia reached by a single research endeavor carried out in Tumkur, India, is surprising. It is probable that the failure to establish an effect that is meaningful is related to the fact that the researchers used a single dose of fenofibrate that was only 5 mg/kg rather than 10 mg/kg in the methodology of that experiment [28].

Most of the newborns who were administered fenofibrate were able to leave the hospital within the first 72 h, whereas the majority of the newborns who were in the control group stayed in the hospital for 96 h. Once more, this conclusion was in agreement with the majority of the investigations carried out all around the world.

## 5. Conclusions

This study reveals that the addition of a single oral dose of 10 mg/kg fenofibrate to conventional phototherapy can lessen the amount of time required to lower the level of bilirubin in a population of children born in the Kurdistan region of northern Iraq. As a direct result of this, the amount of time that these infants spend in the hospital will be reduced. It is abundantly clear that the utilization of fenofibrate as an adjuvant to phototherapy carries with it the potential to significantly improve the administration of care for indirect newborn hyperbilirubinemia. The combination strategy has the potential to improve bilirubin metabolism and hasten the process of removing it from the body more quickly. Before adopting fenofibrate into standard practice, however, it is required to conduct a comprehensive evaluation using well-designed clinical trials in order to ensure the safety and efficacy of this adjuvant medication for the benefit of neonates who suffer from hyperbilirubinemia.

## Figures and Tables

**Figure 1 children-10-01192-f001:**
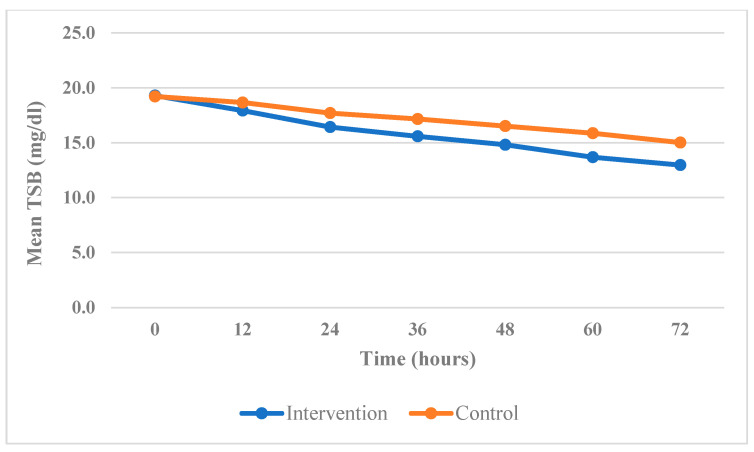
Changes in TSB values with times in the control and intervention groups.

**Table 1 children-10-01192-t001:** Demographic characteristics of the control and intervention groups.

Variables	Intervention Group Mean (SD)	Control Group Mean (SD)	*p*-Value
Age in days	3.74 (1.139)	3.88 (1.239)	0.558 (NS) *
Gestational age in weeks	38.58 (0.758)	38.52 (0.995)	0.735 (NS) *
Weight in kg	3.01 (0.195)	3.018 (0.26)	0.795 (NS) *

* using independent *t*-test, NS: not significant.

**Table 2 children-10-01192-t002:** General and clinical characteristics of the control and intervention groups.

Variables	Intervention Group Mean (SD)	Control Group Mean (SD)	*p*-Value
	No. (%)	No. (%)	
Gender			
Male	28 (56.0)	29 (58.0)	1.00 (NS) **
Female	22 (44.0)	21 (42.0)	
Diagnosis			
ABO incompatibility	25 (50.0)	28 (56.0)	0.862 (NS) **
Rh incompatibility	15 (30.0)	13 (26.0)	
Exaggerated physiological jaundice	10 (20.0)	9 (18.0)	
Direct Coombs test			
Positive	16 (32.0)	13 (26.0)	0.66 (NS) **
Negative	34 (68.0)	37 (74.0)	

** using Chi-squared test, NS: not significant.

**Table 3 children-10-01192-t003:** Mean TSB values in the control and intervention groups at different times of treatment.

Time (Hours)	TSB mg/dL		*p*-Value
	Intervention Group Mean (SD)	Control Group Mean (SD)	
0 (on admission)	19.28 (0.427)	19.21 (0.518)	0.456 (NS)
12	17.92 (0.539)	18.65 (0.573)	0.001
24	16.42 (0.736)	17.69 (0.621)	<0.001
48	14.81 (0.872)	16.51 (0.688)	<0.001
72	12.97 (0.946)	15.01 (1.02)	<0.001
Average time for discharge (h)	63.6 (8.83)	90.9 (8.07)	<0.001

NS: not significant.

## Data Availability

Not applicable.

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
