# Peer review of "Indirect Neonatal Hyperbilirubinemia and the Role of Fenofibrate as an Adjuvant to Phototherapy"

_children, 2023, doi:10.3390/children10071192_

Round 1
Reviewer 1 Report
The study compares the effect of phototherapy and oral Fenofibrate with phototherapy only. The title is wrong and the study must be completely revised.
Line 23: “hours” for “h”
Line 48: “non-pharmacological therapy” for “non pharmacological therapy”
Line 61: “Could” for “coul”
Line 68: space in “of10 mg/kg”
Line 103: Assigning neonates to the study groups is not clear!
Line 126: “The study participants were 100 full term newborns divided into two groups, intervention and control, each of 50”, this statement is reported in line 96
Line 130: “The mean weight was 3.01 Kg and 3.02 Kg in the intervention and control group respectively” for “The mean weight of these neonates in 130 kg was 3.01 and 3.02 in the intervention and control group respectively”
Line 156: “At 24 hours” for “At 24 hr”, use hours every time
Author Response
Dear Reviewer 1,
We have revised all of your comments point-by-point. Please find the attachment...
best regards

Reviewer 2 Report
I have conducted a thorough review of the study titled "The Effect of Oral Fenofibrate on Indirect Neonatal Hyperbilirubinemia". The study aimed to compare the effectiveness of fenofibrate combined with phototherapy versus phototherapy alone in the treatment of neonatal hyperbilirubinemia. I have analyzed the study's objectives, methodology, results, and conclusions, and provide the following review report:
1. Objectives and Significance:
The study addresses an important issue in neonatal care, specifically the treatment of hyperbilirubinemia. Hyperbilirubinemia is a common condition in newborns and effective management is crucial to prevent complications. The objective of the study to evaluate the effectiveness of fenofibrate in reducing bilirubin levels is relevant and significant.
2. Study Design and Methodology:
The study design was a quasi-experimental design with nonrandom allocation of neonates to the intervention and control groups. While a randomized controlled trial would have been preferred, the authors acknowledged the limitations of their design and attempted to ensure comparability between the groups by assessing various characteristics. The use of a single oral dose of 10 mg/kg fenofibrate and the administration of phototherapy in both groups are consistent with previous studies.
3. Results and Findings:
The study findings indicate that the addition of fenofibrate to phototherapy resulted in a significant and earlier reduction in total serum bilirubin (TSB) levels compared to phototherapy alone. The difference in TSB reduction was evident as early as 12 hours after treatment initiation and remained significant at 24, 48, and 72 hours. Most of the neonates in the fenofibrate group were discharged before 72 hours, indicating a shorter hospital stay compared to the control group.
4. Comparison with Previous Studies:
The authors discussed the findings in relation to previous studies conducted in different locations. The early onset of significant TSB reduction observed in this study, compared to other studies, may be attributed to the potentially lower quality of phototherapy in the study setting. Overall, the findings are consistent with previous research that supports the effectiveness of fenofibrate combined with phototherapy in reducing TSB levels in neonatal hyperbilirubinemia.
5. Strengths and Limitations:
The strengths of the study include the focus on a clinically relevant topic, a well-defined intervention, and a systematic assessment of various characteristics to ensure comparability between the groups. However, the study has certain limitations, including the quasi-experimental design, nonrandom allocation of participants, and potential variations in the quality of phototherapy. These limitations should be considered when interpreting the results.
6. Conclusion:
The study provides evidence that adding a single oral dose of 10 mg/kg fenofibrate to conventional phototherapy can effectively reduce bilirubin levels in neonates with hyperbilirubinemia. The earlier and significant reduction in TSB levels, as well as the shorter hospital stay observed in the intervention group, suggest that this intervention has the potential to improve neonatal care and reduce healthcare costs.
In summary, the study contributes valuable insights into the use of fenofibrate in the management of neonatal hyperbilirubinemia. Despite some limitations, the findings support the effectiveness of fenofibrate combined with phototherapy in reducing bilirubin levels and shortening hospital stays. Further research, particularly randomized controlled trials, would be beneficial to confirm these findings and explore potential variations in response to fenofibrate.
The quality of English language used is generally good. The sentences are coherent, and the ideas are conveyed clearly. The study follows the standard structure and style of scientific writing, including the use of appropriate terminology and academic language.
However, it's important to note that the given excerpts are limited, and a comprehensive assessment of the entire study would be necessary to provide a complete evaluation of the English language quality. Additionally, without having access to the full study, it is difficult to assess the consistency and overall proficiency of the writing.
Author Response
Dear Reviewer 2,
We have revised all of your comments point-by-point. Please find the attachment...
best regards

Round 2
Reviewer 1 Report
I can't open the review response file
Author Response
Dear Reviewer 1,
We have revised all of your comments and grammatical mistakes again and tracked them with the color yellow. Please find the attachment...
best regards

Round 3
Reviewer 1 Report
Now it is better